# Safety of Monkeypox Vaccine Using Active Surveillance, Two-Center Observational Study in Italy

**DOI:** 10.3390/vaccines11071163

**Published:** 2023-06-27

**Authors:** Marco Montalti, Zeno Di Valerio, Raffaella Angelini, Elena Bovolenta, Federica Castellazzi, Marta Cleva, Paolo Pandolfi, Chiara Reali, Davide Resi, Renato Todeschini, Davide Gori

**Affiliations:** 1Unit of Hygiene, Department of Biomedical and Neuromotor Sciences, Public Health and Medical Statistics, University of Bologna, Via San Giacomo 12, 40126 Bologna, Italy; marco.montalti7@studio.unibo.it (M.M.); davide.gori4@unibo.it (D.G.); 2Department of Public Health, Romagna Local Health Authority, Via della Rocca 19, 47121 Forlì, Italy; 3Department of Public Health, Bologna Local Health Authority, Via Gramsci 12, 40121 Bologna, Italy

**Keywords:** monkeypox, AEFIs, adverse events, vaccine surveillance, vaccine preventable diseases, MSMs, pharmacovigilance, Imvanex, Jynneos

## Abstract

In August 2022, Italy launched a vaccination campaign to combat the spread of the monkeypox virus, which the WHO has designated as a public health emergency. Priority targets for the campaign included laboratory personnel and men who have sex with men with specific risk criteria. Primary immunization involved two doses of the Imvanex/Jynneos vaccine, followed by a single booster dose. We conducted a study in two Italian towns, Bologna and Forlì, in October and November 2022 to investigate adverse events following immunization (AEFIs) of the monkeypox vaccine through participant-based active surveillance. Participants who received the vaccine and were aged 18 and over were invited to complete an e-questionnaire by scanning a QR code during their second vaccine appointment or by email one month after the booster dose. A descriptive analysis of AEFI incidences was conducted, with the results stratified by type and severity of symptoms. A total of 135 first-dose, 50 second-dose, and 6 single-dose recipients were included, with a mean age of 36.4 ± 8.7 years. Systemic reactions after the first and second doses were reported by 39.3% and 26.0% of participants, respectively, with asthenia being the most common symptom. Local site reactions were reported by 97.0% and 100.0% of participants, respectively, with redness, swelling, and itching being the most common local AEFIs. Grade 3 or 4 AEFIs were reported for local AEFIs only by 16.8% and 14.0% of participants after the first and second doses, respectively. Our findings suggest that the monkeypox vaccine has a high tolerability profile in terms of short-term common systemic AEFIs. However, the high incidence and severity of local AEFIs highlight the need to monitor their persistence following intradermal administration of the vaccine.

## 1. Introduction

Monkeypox is a zoonotic infectious disease caused by the Monkeypox virus (MPXV) [1], which causes a disease similar in certain ways to smallpox, and the respective infections appear to be immunologically cross-reactive and cross-protective with each other [2]. MPXV infection, first sporadic and confined to territories in Central and West Africa, has, since May 2022, rapidly spread to countries not typically endemic globally, in the absence of epidemiological links to endemic areas, and with transmission and clinical features atypical of previous outbreaks, spreading predominantly within the GBMSM community (gay, bisexual, transgender, and other men who have sex with men) [3,4].

Given the exponential increase in cases, the World Health Organization (WHO) declared monkeypox a “Public Health Emergency of International Concern” (PHEIC) on 23 July 2022 [5].

Both the European Centre for Disease Prevention and Control (ECDC) and WHO have launched a series of public health measures aimed at stemming the ongoing epidemic, such as timely identification, management (including isolation), and reporting of cases, as well as contact tracing. In addition, a vaccination strategy has been developed targeting those groups most at risk [6,7].

It is, to date, the most concerning Orthopoxvirus infection since the eradication of smallpox (1980) [1]. Its sudden spread is probably related to the weakening of herd immunity resulting from the smallpox vaccination, which provided partial protection, and the increase in the susceptible naïve population as a result of the discontinuation of the vaccination program [8,9]. Until 9 February 2023, a total number of 85,765 cases, including 93 deaths, were reported to the WHO, distributed in 110 states globally [10]. Of these, in 96.6% of cases, the affected individuals were male with a mean age of 34 years (IQR: 29–41), and in 48.1% of cases, the affected individuals were HIV-positive. Among cases whose sexual orientation was known, 84.1% identified themselves as men who have sex with men (MSM) [10].

The first case in Italy was reported on 20 May 2022, and until 7 February 2023, a total of 955 cases were confirmed, 252 of which were travel-related abroad [11]. The average age of the predominantly affected group is 37 years (IQR: 14–71), and 943 cases have been reported of male subjects. The regions with the highest absolute number of cases are represented by Lombardy (410), Lazio (161), and Emilia-Romagna (89) [11].

In Italy, a Ministry of Health guidance was issued in early August 2022 containing recommendations and protocols to be followed in order to confine the spread of MPXV infection, aimed at timely identification, management (including isolation), reporting of cases, and contact tracing [12]. Finally, considering the decisive role of the vaccination in interrupting the chain of transmission with 85% protection against infection [13], a specific vaccination campaign started in August 2022 with the first doses delivered to the Italian regions most affected by the infection [14].

Given the limited availability of doses, the vaccination was offered to laboratory personnel with possible direct exposure to the virus and to GBMSM who fit additional specific risk criteria: having a recent history of multiple sexual partners; participating in group sex events; engaging in sexual encounters in clubs, cruising, or saunas; having recently acquired a sexually transmitted infection; and participating in chemsex (combination of drug use and sexual activity) [14].

The vaccine used was Imvanex/Jynneos, which safety profile was determined with 22 clinical trials conducted in 7800 subjects, both naïve and previously immunized [15]. As reported in the Jynneos datasheet revised by the Food and Drug Administration (FDA) in April 2022, in the healthy, smallpox vaccine-naïve population, for any dose of vaccine administered, the most commonly encountered local AEFIs were pain (84.9%), redness (60.8%), swelling (51.6%), induration (45.4%), and itching (43.1%) at the injection site [15].

In contrast, muscle pain (42.8%), headache (34.8%), asthenia (30.4%), nausea (17.3%), and chills (10.4%) were among the most frequently elicited systemic AEFIs. A febrile upsurge was found in only 1.7% of subjects [15].

The clinical trials preceding vaccine approval provided crucial evidence, but they were not enough on their own. It is important to gather real-world data alongside the vaccine campaign to ensure ongoing safety. Vaccine pharmacovigilance, which involves the post-marketing monitoring of vaccines, is essential in this regard. It is defined as the science and activities related to detecting, assessing, understanding, and communicating adverse events following immunization (AEFIs) and other issues related to vaccines or immunization, as well as preventing any negative effects that may arise from a vaccination [16]. Even if the MPXV outbreak has been relatively limited in the number of cases and severity, a continued assessment of the vaccines against the Orthopoxvirus genus will remain relevant as long as re-emergence and spillover events are a possibility.

The purpose of this study was to carry out an active vaccine surveillance program (i.e., solicited reporting) and present information about the frequency and characteristics of AEFIs related to the monkeypox vaccine in the Emilia Romagna region of Italy, specifically the cities of Bologna and Forlì. Additionally, we intended to address and analyze specifically the differences between the first and the second doses.

## 2. Materials and Methods

### 2.1. Study Design and Participants

This was an active pharmacovigilance study to assess the safety profile of the monkeypox vaccine in Bologna and Forlì (Emilia-Romagna, Italy). Both cities had only one vaccination center offering the vaccination. The vaccination process included, for primary immunization, two doses of Imvanex/Jynneos (0.1 mL) administered intradermally on day 0 and after day 28. People who had been vaccinated for smallpox in the past received only one booster intradermal dose (0.1 mL). Physicians and public health officers at both vaccination centers in Bologna and Forlì recruited participants for the study. All individuals who received the vaccine were actively recruited following the administration of their second or booster doses. The eligibility criterion for the study was established as having received at least one dose of the monkeypox vaccine. The exclusion criterion for the study included individuals who were unable to understand or respond to the questionnaires accurately. There was no randomization or specific selection process conducted for participation. All participants provided informed consent to be included in the study database.

The study protocol was reviewed and approved by the University of Bologna Ethics Committee with approval number 0366819 in December 2022. This study followed the Strengthening the Reporting of Observational Studies in Epidemiology (STROBE) criteria.

### 2.2. Outcomes

The primary outcome of this study was to measure the safety of the monkeypox vaccine by collecting patient-reported data on AEFIs in the first week/month after the first and second doses. This allowed for real-time monitoring of the safety data and the calculation of the incident rates of AEFIs. Another objective was to compare the occurrence of AEFIs between the first and second doses, both quantitatively and qualitatively.

### 2.3. Procedure and Questionnaire

Participants were actively given standardized e-questionnaires to self-report any potential AEFIs. The questionnaires were generated using Google Forms and could be filled out independently or with the assistance of staff members.

All vaccine recipients in both centers were asked to fill in the questionnaires referring to their first-dose AEFIs during the administration of their second dose (Q1), with a questionnaire acceptance rate of 48%. Questionnaires referring to the second dose (Q2) and booster doses (Q3) were filled out one month after the injection. The questionnaires investigated demographic information (age, weight, height, gender, and profession); anamnestic data (diseases, therapy, vaccine coadministrations, and allergies); the potential AEFIs occurring in the week/month after the first dose (Q1) and the booster dose (Q3); and the severity/impact of the symptoms (including the need for medical assistance and hospitalization). Q2 investigated the potential AEFIs occurring in the first week/month after the second dose. To link the questionnaires to one another, email addresses were asked in all the questionnaires. To minimize missing data, participants were required to provide answers to questions regarding relevant variables in the questionnaire. The list of AEFIs and most of the questions in the e-questionnaires were based on vaccine surveillance studies conducted by the European Medicines Agency. The AEFIs indicated were consistent with those typically identified by relevant regulatory agencies and the expected latency of occurrence. The questionnaires specified the clinical characteristics, frequency, and severity of any AEFIs using the Common Terminology Criteria for Adverse Events (CTCAE) version 5.0 scale. Grade 1 included asymptomatic or mild symptoms that did not require intervention, and moderate symptoms were classified as Grade 2, while severe or medically significant symptoms were classified as Grade 3. Symptoms that required urgent intervention were classified as Grade 4.

Each participant was provided with an information sheet and an informed consent form attached to the e-questionnaire, and they consented to participate in the survey after reviewing the provided information. To ensure the privacy of participants, any sensitive information collected was stored anonymously.

### 2.4. Data Collection and Analysis

The timeframe for patient recruitment lasted from 1 October to 30 November 2022. The collection timeframe for the data included in this analysis was from 1 October to 30 December 2022. Data collection was carried out through the administration of a questionnaire with two different approaches: QR-code scanning at vaccination sites for second-dose recipients only (referring to their first vaccine dose) and through a link sent via e-mail for booster doses and second-dose recipients. The mean and standard deviation were used to summarize numerical variables, while frequencies and percentages were used to summarize categorical variables. All analyses were carried out using Stata software, version 17 (StataCorp, 2021, Stata Statistical Software: Release 17, College Station, TX, USA: StataCorp LP).

## 3. Results

### 3.1. Population Characteristics

We recruited 141 male participants who received the monkeypox vaccine between October and November 2022. The mean age was 36.4 ± 8.7 years, and the most represented age group was 25–34 years (41.8%), followed by 35–44 years (31.9%) (Table 1).

Only 3.6% of the study participants had a BMI > 30. Almost eighty percent of the respondents (79.4%) were healthy and did not report being affected by pre-existing diseases.

At the time of vaccination, 42.6% of participants were taking medications, including psychotropic drugs (5.0%), pain medications (3.6%), antihypertensives (1.4%), antihistamines (1.4%), and corticosteroids (1.4%). As many as 27.7% of patients were on antivirals.

As many as 65.3% of patients reported no allergies. Among the allergic patients (34.8%), we found conditions such as rhinitis (22.0%), drug allergies (9.2%), asthma (4.3%), contact dermatitis/urticaria (1.4%), food allergies (0.7%), and insect allergies (0.7%). 

Only 16.3% of respondents reported being healthcare workers. Among them, the most represented professions were physicians (6.4%), pharmacists (1.4%), volunteers (1.4%), laboratory technicians (0.7%), and others (4.3%) (Table 1).

### 3.2. Administration Context and AEFIs

The majority of the participants reported taking no medications either prior to the administration of the first dose (91.9%) or at the administration of the second dose (88.0%). Anti-inflammatories and antihistamines were taken before the first dose in 4.4% and 3.7% of cases, respectively, and before the second dose in 10.0% and 2.0% of cases, respectively. Only 20.0% of participants received a concomitant vaccination with the first dose and 34.0% with the second dose (Table 2).

Local adverse reactions were reported by 97% of patients within the first week after administration of the first dose. The main local AEFIs reported by this category were redness (94.1%), swelling (84.4%), and itching (79.3%) at the injection site. All second-dose recipients experienced the onset of local adverse reactions within the first week after vaccination. The most frequently reported local AEFIs from this category were redness (96.0%), swelling (84.0%), and itching (74.0%) at the injection site (Table 2).

Systemic AEFIs were reported by 39.3% of first-dose recipients. Within the first week after vaccination, asthenia (29.6%) and headaches (16.3%) mainly occurred. These symptoms were also experienced by 40.0% of the second-dose recipients. Patients mainly reported the occurrence of asthenia (20.0%), headaches (12.0%), and malaise (12.0%) arising within the first week after administration (Table 2).

Local and/or systemic AEFIs usually appeared within two days (45.2%) after the first dose. Less frequently, signs and symptoms appeared within seconds/minutes (12.6%) or in one week (10.4%) after vaccination. Only 26.7% of participants reported the onset of some local and/or systemic AEFIs more than one week after the first dose. These adverse reactions resolved in more than one week (46.7%) or within one week (16.3%) after the first dose. Some (21.5%) of first-dose recipients were still symptomatic at the time of the second-dose injection. The second-dose AEFIs similarly appeared within two days (52.0%) or within seconds/minutes (26.0%). More rarely, AEFIs appeared after one week (12.0%) (Table 2).

### 3.3. AEFI Grading

The local first-dose AEFI severity was typically mild/moderate: Grade 2 intensity in 63.4% of cases and Grade 1 in 19.8% of cases. Clinically severe reactions or those requiring medical intervention were reported by 16.8% of patients: 6.9% of local AEFIs were Grade 3 and 9.9% Grade 4. The local AEFIs following the second dose were mostly mild/moderate: Grade 2 in 66.0% of cases and Grade 1 in 20.0% of cases. Severe Grade 3 and Grade 4 local AEFIs were found in only 10.0% and 4.0% of cases, respectively (Table 2 and Figure 1).

Seventy percent of first-dose systemic AEFIs were Grade 1, and 27.5% were Grade 2. Only two percent experienced severe systemic AEFIs that required medical intervention (Grade 4). Additionally, the systemic second-dose AEFIs were mostly mild/moderate: Grade 1 in 53.8% of cases and Grade 2 in 38.5% of cases. Among these, only 7.7% of patients experienced a severe systemic AEFI (Grade 3) (Table 2 and Figure 1).

Finally, only among recipients of both doses (*n* = 50), the percentage of participants who reported the occurrence of local and/or systemic AEFIs for both the first and second doses was assessed (Table 3). Those who experienced local first-dose AEFIs tended to report them following the second dose as well (92.0%). This was not as true for systemic AEFIs: 22% of participants presented only first-dose systemic AEFIs but not following the second dose, 20% only second-dose AEFIs, and 20% both first- and second-dose AEFIs.

### 3.4. Single-Dose Recipients

A total of six respondents received a single dose as a booster. Their ages ranged from 51 to 62 years (mean: 56 ± 3.95). All of them reported at least one local AEFI, of which two were at least Grade 3 or 4. None reported systemic AEFIs.

## 4. Discussion

This study assessed the safety of the monkeypox vaccine in a real-world context of active surveillance, finding high rates of local, albeit mild/moderate, reactions and modest rates of systemic reactions following intradermal administration of the vaccine.

In line with the Ministry of Health guidelines for access to the vaccination [14], participants who took part in the study were considered at higher risk for contracting monkeypox, namely GBMSM and some laboratory personnel. Interestingly, 27.7% of the survey sample was on antiretroviral therapy, and 6.4% of the vaccine recipients reported being immunocompromised. While the cause of their immunosuppressed status has not been investigated in detail, these data could be explained by a significant presence of people living with HIV or on pre-exposure prophylaxis at the vaccination centers.

The type and frequency of local AEFIs reported mostly overlapped between the first- and second-dose vaccine recipients. Redness, swelling, and itching at the injection site represented the most frequent local AEFIs: 94.1–96.0%, 84.4–84.0%, and 79.3–74.0% following the first and second doses, respectively. The reported incidence rates of local AEFIs were higher than those reported in the vaccine datasheet [15] and, also, for other vaccines [17,18,19]. However, the intradermal administration could have an impact on increasing the local AEFIs frequencies if compared to the intramuscular or subcutaneous routes [20,21].

Systemic AEFIs, which were much less frequent than local ones, in the order of occurrence, were asthenia, headaches, generalized malaise, and myalgia. Common and very common systemic AEFIs were generally less frequent than those reported in the vaccine datasheet [15] and generally did not differ in frequency from those found with other currently available vaccinations [17,18,19].

For both local and systemic AEFIs, mostly Grade 1 and 2 reactions were reported. Only two occurrences of Grade 3 and 4 systemic AEFIs (requiring medical attention) were registered. First-dose local AEFIs occurred in 6.9% of cases that were Grade 3 and in 9.9% of cases that were Grade 4. Second-dose local AEFIs occurred in 10.0% of cases that were Grade 3 and in 4.0% of cases that were Grade 4. It was likely that the persistence and severity of the local reactions led to the seeking out of medical consultations and the use of topical medication. Finally, no unexpected and/or serious adverse reactions or deaths attributable to the vaccination were observed.

In our study, the latency time window for the onset of local and/or systemic AEFIs was comparable after the administration of the first and second doses, with most AEFIs appearing two days after vaccination. Regarding the timing of resolution, among first-dose recipients, the symptoms usually resolved more than a week after vaccination. Interestingly, however, 21.5% of patients were still symptomatic one month after the injection. Indeed, the time window required for the resolution of the symptoms found was longer than that observed during clinical trials (more than a week vs. 1–6 days) [12]. 

Overall, the frequency and type of local and systemic AEFIs were similar between the first and second doses. We observed that those who experienced local AEFIs following the first dose had a high tendency to report them following the second dose as well. This was not as true for systemic AEFIs.

From May to October 2022, the Center for Disease Control (CDC) monitored the monkeypox vaccine safety. Even if this was passive surveillance and the data collected were difficult to compare because of the different types of symptoms investigated, it was interesting to note that the CDC results seemed to confirm both the increase in local AEFIs following intradermal administration of the vaccine (particularly erythema at the injection site and urticaria) and the mild/moderate grade of the symptoms [22].

In a phase 3 randomized, double-blinded study conducted in the United States in 2022 among 1129 patients, very frequent local AEFIs (91.2%) were reported in line with our study, albeit with more massive pain at the injection site reported [23]. In the same study, a higher frequency of systemic AEFIs was found, albeit of mild/moderate grades, as in our study [23]. In addition to the different routes of administration (intradermal in our study and subcutaneous in the U.S. study), one possible explanation for some of the differences in AEFIs frequencies could be that our study population was highly selected (males with a mean age of 36 years).

In a very recent research letter published in *JAMA* describing the short-term AEFIs of the monkeypox vaccine in more than 20,000 vaccine recipients, our findings on the higher occurrence of local AEFIs compared with systemic (especially via intradermal administration) were confirmed [24]. However, in our study and previous studies, the absolute rates of events were higher than in the study by Deng et al., which reported 21% systemic AEFIs and 52% local AEFIs. The good systemic tolerability of the vaccine was also confirmed. In fact, there was a low percentage of people in the Deng et al. study who reported medical appointments or missed daily activities [24].

Although active vaccine vigilance studies overcome the underreporting phenomenon associated with the spontaneous reporting system (passive surveillance), they present some limitations, attributable mainly to the size and characteristics of the sample. The latter, in our study, turned out to be restricted access to the vaccination, which was allowed only to selected targets. Another important limitation is that self-reporting surveys may compromise the validity and reliability of the data collected, especially due to biases related to social desirability, exaggeration or underestimation of data, mistakes, or misremembering. Another major limitation of the study was determined by the presence of coadministration, with 20% of the first-dose recipients and 34.0% of the second-dose recipients given another vaccine concurrently. In this not-insignificant proportion of patients, the direct cause of the AEFIs could not be tracked. Although the list of AEFIs reported on the questionnaire was adapted from EMA surveillance studies, some rare or unusual reactions might have been missed. In fact, it should be mentioned that 8.4% of participants reported the occurrence of AEFIs not specified on the questionnaire (listed as “others”). Finally, the follow-up time window was limited to one month after the vaccination; any AEFI that occurred long afterward was not detected.

## 5. Conclusions

This active surveillance study allowed us to confirm the monkeypox vaccine’s common and very common short-term AEFI good safety profiles. Overall, we found no major differences in the type and occurrence of first- and second-dose AEFIs. The vaccine has a high tolerability profile, especially in terms of common short-term systemic AEFIs. Although a high rate of common local AEFIs was found, these were mostly mild/moderate. The high incidence of signs and symptoms such as redness, swelling, and itching at the injection site, however, underlines the need to monitor for the possible persistence of local AEFIs following intradermal administration.

## Figures and Tables

**Figure 1 vaccines-11-01163-f001:**
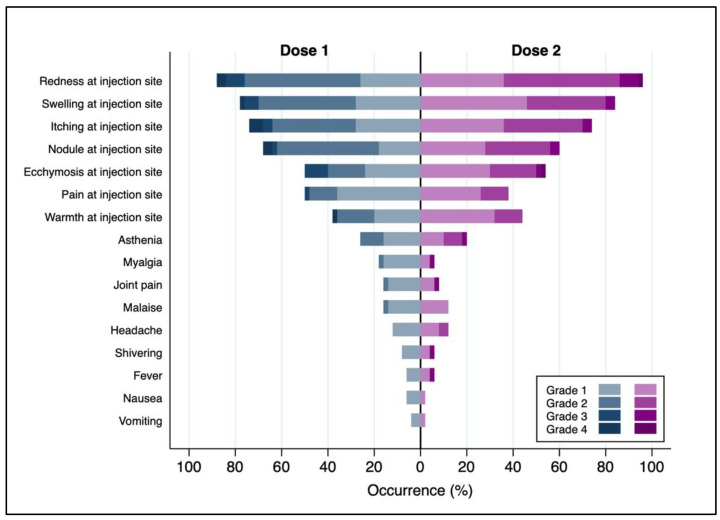
Severity and occurrence of first- and second-dose local and systemic AEFIs.

**Table 1 vaccines-11-01163-t001:** Main sample characteristics (N = 141).

Characteristics			N (%)
Age, years (mean (SD))			36.4 (8.7)
BMI > 30	No		136 (96.5)
	Yes		5 (3.6)
Preexisting health conditions	No		112 (79.4)
	Yes		29 (20.6)
		Immunodepression	9 (6.4)
		Psychiatric conditions	6 (4.3)
		Heart disease	3 (2.1)
		Bone and joint disease	2 (1.4)
		Diabetes	1 (0.7)
		Hypertension	1 (0.7)
		Liver disease	1 (0.7)
		Kidney disease	1 (0.7)
		Cancer	1 (0.7)
		Other	7 (5)
Drugs	No		81 (57.5)
	Yes		60 (42.6)
		Antiviral	39 (27.7)
		Psychiatric drugs	7 (5)
		Pain medication	5 (3.6)
		Anti-hypertensive	2 (1.4)
		Antihistamines	2 (1.4)
		Corticosteroids	2 (1.4)
		Other	6 (4.3)
Allergies	No		92 (65.3)
	Yes		49 (34.8)
		Rhinitis	31 (22)
		Drug allergy	13 (9.2)
		Asthma	6 (4.3)
		Contact dermatitis/Hives	2 (1.4)
		Food allergy	1 (0.7)
		Insects	1 (0.7)
		Other	8 (5.7)
Healthcare worker	No		118 (83.7)
	Yes		23 (16.3)
		Medical Doctor	9 (6.4)
		Other healthcare worker	9 (6.4)
		Pharmacist	2 (1.4)
		Volunteer	2 (1.4)
		Laboratory technician	1 (0.7)

**Table 2 vaccines-11-01163-t002:** Administration context and first- and second-dose AEFIs.

			First Dose (*n* = 135)	Second Dose (*n* = 50)
Did you take one or more drugs before the vaccine?	No		124 (91.9)	44 (88.0)
	Yes		11 (8.1)	6 (12.0)
		Antiinflammatory drugs	6 (4.4)	5 (10.0)
		Antihistamines	5 (3.7)	1 (2.0)
		Acetaminophen	0 (0.0)	0 (0.0)
Were other vaccines administered at the same time as the vaccine? ^c^	No		108 (80.0)	33 (66.0)
	Yes		27 (20.0)	17 (34.0)
Did you experience any AEFI within the first week after the vaccine?	No		1 (0.7)	0 (0.0)
	Yes		134 (99.3)	50 (100.0)
		Local AEFIs	131 (97.0)	50 (100.0)
		Redness at injection site	127 (94.1)	48 (96.0)
		Swelling at injection site	114 (84.4)	42 (84.0)
		Itching at injection site	107 (79.3)	37 (74.0)
		Nodule at injection site	97 (71.9)	30 (60.0)
		Ecchymosis at injection site	69 (51.1)	19 (38.0)
		Pain at injection site	62 (45.9)	27 (54.0)
		Warmth at injection site	61 (45.2)	22 (44.0)
		Systemic AEFIs	53 (39.3)	13 (26.0)
		Asthenia	40 (29.6)	10 (20.0)
		Headache	22 (16.3)	3 (6.0)
		Malaise	20 (14.8)	4 (8.0)
		Myalgia	19 (14.1)	6 (12.0)
		Joint pain	16 (11.9)	6 (12.0)
		Shivering	9 (6.7)	3 (6.0)
		Nausea	7 (5.2)	3 (6.0)
		Fever	6 (4.4)	1 (2.0)
		Vomiting	4 (3.0)	1 (2.0)
		Other	6 (4.4)	2 (4.0)
Within how long after the vaccination did symptoms appear?		No AEFIs	1 (0.7)	4 (8.0)
		Seconds/minutes	17 (12.6)	13 (26.0)
		Two days	61 (45.2)	26 (52.0)
		One week	14 (10.4)	6 (12.0)
		Does not recall	6 (4.4)	1 (2.0)
		No answer	36 (26.7)	0 (0.0)
Did any adverse effects appear later than one week after the vaccination?		No	99 (73.3)	42 (84.0)
		Yes	36 (26.7)	8 (16.0)
After how long did the AEFI disappear? ^a^		No AEFIs	1 (0.7)	n.a.
		Within one week	22 (16.3)	n.a.
		Later than one week	63 (46.7)	n.a.
		Still present	29 (21.5)	n.a.
		No answer	20 (14.8)	n.a.
Reported maximum severity of AEFIs ^b^	Local AEFIs	Grade 1	26 (19.8)	10 (20.0)
		Grade 2	83 (63.4)	33 (66.0)
		Grade 3	9 (6.9)	5 (10.0)
		Grade 4	13 (9.9)	2 (4.0)
	Systemic AEFIs	Grade 1	36 (70.6)	7 (53.8)
		Grade 2	14 (27.5)	5 (38.5)
		Grade 3	0 (0)	1 (7.7)
		Grade 4	1 (2)	0 (0)

^a^ Question not included in the second questionnaire. ^b^ Highest grade reaction reported. ^c^ The other vaccinations administered by risk category, health condition, or age were mainly Hepatitis A, Papillomavirus, and Tetanus.

**Table 3 vaccines-11-01163-t003:** Reporting of first- and second-dose AEFIs among the respondents to both Q1 and Q2 (*n* = 50).

		Local second-dose AEFIs		
		No	Yes	Total
Local first-dose AEFIs	No	0 (0.0)	3 (6.0)	3 (6.0)
	Yes	1 (2.0)	46 (92.0)	47 (94.0)
	Total	1 (2.0)	49 (98.0)	50 (100.0)
		Systemic second-dose AEFIs		
Systemic first-dose AEFIs	No	19 (38.0)	10 (20.0)	29 (58.0)
	Yes	11 (22.0)	10 (20.0)	21 (42.0)
	Total	30 (60.0)	20 (40.0)	50 (100.0)
		Any second-dose AEFIs		
Any first-dose AEFI	No	0 (0.0)	1 (2.0)	1 (2.0)
	Yes	1 (2.0)	48 (96.0)	49 (98.0)
	Total	1 (2.0)	49 (98.0)	50 (100.0)

## Data Availability

Data supporting the reported results can be obtained from the corresponding author.

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
