# Peer review of "Safety of Monkeypox Vaccine Using Active Surveillance, Two-Center Observational Study in Italy"

_vaccines, 2023, doi:10.3390/vaccines11071163_

Round 1

Reviewer 1 Report

The paper is well-designed and provides good information.

Author Response

Rev 1

The paper is well-designed and provides good information.

Authors

We thank the Reviewer for the appreciation of our work and the time spent on their review.

Reviewer 2 Report

Thank you for giving me the opportunity to review this manuscript. The work is timely based on the recent public health emergency declared by the WHO. I have few comments as follows:

1. The authors need to declare that reported AEFIs are self-reported by the respondents and not based on diagnostic clinical data. This could be mentioned in the limitations part.

2. How could do authors describe potential confounding factors in the limitations part when no inferences were made in the results? The results was descriptive in nature? I would expect to see some form of causal inference in this circumstances to detect confounders or mediators. Probably a DAG diagram would do?

3. The authors would need to mention the likely schedule of the immunization program, when would dose 2 and booster doses would be administered, and what would be the criteria to warrant a booster?

4. In Table 1, the authors would need to put a foot note on what constitutes "Others" in the variables for pre-existing conditions, drugs, allergies, other healthcare worker?

5. The manuscript is substantially limited with missing strategies for sampling technique and selection processes. Authors were dealing with vulnerable groups, didn't authors consider snowball sampling as a potential strategy to recruit a wider range of samples in both towns rather than recruiting participants who visited the vaccination centers only? 

6. The discussion lacks plausibility on the implications for clinical and public health practice.

English language is fine, some minor corrections needed.

Author Response

Rev2
Thank you for giving me the opportunity to review this manuscript. The work is timely based on the recent public health emergency declared by the WHO. I have few comments as follows:

  1. The authors need to declare that reported AEFIs are self-reported by the respondents and not based on diagnostic clinical data. This could be mentioned in the limitations part.

Authors

We thank the Reviewer for this comment, which allowed us to better specify in the methods the fact that AEFIs were self-reported and to discuss this limitation in the relevant section.

2. How could do authors describe potential confounding factors in the limitations part when no inferences were made in the results? The results was descriptive in nature? I would expect to see some form of causal inference in this circumstances to detect confounders or mediators. Probably a DAG diagram would do?

Authors

We thank the reviewer for this comment and have modified the study limits accordingly. Indeed, the findings are only descriptive in nature: this is due to the small sample size. We conducted a multivariate analysis and included variables selected through a mixed method of stepwise regression and predictors of AEFIs confirmed by literature to check for possible associations in the results, but as expected they were not statistically significant. We think that further analysis, conducted on a larger scale, would certainly help to study the phenomenon.

  1. The authors would need to mention the likely schedule of the immunization program, when would dose 2 and booster doses would be administered, and what would be the criteria to warrant a booster?

Authors

The vaccine schedule and criteria for the booster dose are described in the first lines of the methods section.

  1. In Table 1, the authors would need to put a foot note on what constitutes "Others" in the variables for pre-existing conditions, drugs, allergies, other healthcare worker?

Authors

We thank the reviewer for this comment. The variable "other" is not an aggregate variable, but it was like that in the questionnaire, which did not include the option to detail further.

  1. The manuscript is substantially limited with missing strategies for sampling technique and selection processes. Authors were dealing with vulnerable groups, didn't authors consider snowball sampling as a potential strategy to recruit a wider range of samples in both towns rather than recruiting participants who visited the vaccination centers only?

Authors

We thank the reviewer for this comment that allows us to better clarify the selection process. The only people vaccinated for MPX in the two cities were vaccinated in the two vaccine centers that joined the study because these were the only two centers that had vaccine availability. For these reasons, the snowball technique would not have been necessary, having described the study and proposed recruitment to all recruitable subjects.

We have now specified this issue in more detail in the Methods section.

  1. The discussion lacks plausibility on the implications for clinical and public health practice.

Authors

We understand the Reviewer's point of view even if we do not completely agree with them. We think that commenting on vaccine vigilance and active monitoring of AEFIs in the light of other studies published on the same vaccine or comparing them to other commonly used vaccinations is of great use.

Observing how frequencies of vaccine specific AEFIs following first and second doses are similar, local AEFIs following intradermal administration are particularly frequent, and well-tolerated systemic AEFIs could be all useful elements for both clinicians and public health professionals.

Reviewer 3 Report

Thank you for the opportunity to review this important paper describing AEFIs reported through active surveillance following administration of the monkeypox vaccine in 141 male patients in a region of Italy. This paper adds to the literature and expands our understanding of the safety of the MPXV vaccine.

The following comments are offered to improve the paper.

TITLE: suggest referring to vaccine by generic name rather than brand.

ABSTRACT: define the abbreviation "MSM". Define the abbreviation "MPXV"

Body of paper: refer to the brand name of the vaccine once in the methods section of the manuscript. Otherwise, refer to the vaccine by generic name.

INTRODUCTION:

Lines 59-60, change "men who have relationships with men (MSM) to "men who have sex with men (MSM)". 

Line 61-62 "The first case in Italy..." add citation for this statement.

Line 66 "In the country, a Ministry of Health guidance..." Specify country. Italy?

Line 77 define the term "chemsex" (add a brief definition in parentheses next to the term). 

Lines 80-83 "As reported in the Jynneos datasheet revised by the Food and Drug Administration..." add citation for this statement.

METHODS 2.4 Data collection and analysis - consider adding an analysis to evaluate the effect of other vaccines administered at the same time as the monkeypox vaccination. Was the severity of local or systemic AEFIs associated with receiving more than 1 vaccine?

RESULTS

Lines 179 and 180: please describe the response rate to the e-questionnaires. How many patients were recruited versus responded? 

Table 2. elaborate on the other vaccines administered at the same time as the MPXV vaccine. List the other vaccines in the table.

DISCUSSION:

Consider adding the following citation and study results into the discussion section of your paper. 

Deng L, Lopez LK, Glover C, Cashman P, Reynolds R, Macartney K, Wood N. Short-term Adverse Events Following Immunization With Modified Vaccinia Ankara-Bavarian Nordic (MVA-BN) Vaccine for Mpox. JAMA. 2023 May 5. doi: 10.1001/jama.2023.7683. Epub ahead of print. PMID: 37145654.

Author Response

Rev3

Thank you for the opportunity to review this important paper describing AEFIs reported through active surveillance following administration of the monkeypox vaccine in 141 male patients in a region of Italy. This paper adds to the literature and expands our understanding of the safety of the MPXV vaccine.

Authors

We thank the Reviewer for the appreciation of our work and the valuable time spent on their review.

The following comments are offered to improve the paper.

TITLE: suggest referring to vaccine by generic name rather than brand.

Authors

The manuscript has been modified accordingly.

ABSTRACT: define the abbreviation "MSM". Define the abbreviation "MPXV"

Authors

The manuscript has been modified accordingly.

Body of paper: refer to the brand name of the vaccine once in the methods section of the manuscript. Otherwise, refer to the vaccine by generic name.

Authors

The manuscript has been modified accordingly.

INTRODUCTION:

Lines 59-60, change "men who have relationships with men (MSM) to "men who have sex with men (MSM)".

Authors

The manuscript has been modified accordingly.

Line 61-62 "The first case in Italy..." add citation for this statement.

Authors

Citation added.

Line 66 "In the country, a Ministry of Health guidance..." Specify country. Italy?

Authors

The manuscript has been modified accordingly.

Line 77 define the term "chemsex" (add a brief definition in parentheses next to the term).

Authors

Brief definition added.

Lines 80-83 "As reported in the Jynneos datasheet revised by the Food and Drug Administration..." add citation for this statement.

Authors

Citation added.

METHODS 2.4 Data collection and analysis - consider adding an analysis to evaluate the effect of other vaccines administered at the same time as the monkeypox vaccination. Was the severity of local or systemic AEFIs associated with receiving more than 1 vaccine?

Authors

We thank the reviewer for this comment and have better clarified the study limits accordingly. Indeed, the findings are only descriptive in nature, even given the small sample size. We have conducted a multivariate analysis with covariates selected through a stepwise process and by including known predictors of AEFIs to check for possible associations in the results, but there were no statistically significant findings (not even between AEFIs severity and receiving more than 1 vaccine).
We think that further analysis, conducted on a larger scale, would certainly help to study the phenomenon.

RESULTS

Lines 179 and 180: please describe the response rate to the e-questionnaires. How many patients were recruited versus responded?

Authors

We thank the reviewer for this suggestion. We have included the acceptance rate in the manuscript.

Table 2. elaborate on the other vaccines administered at the same time as the MPXV vaccine. List the other vaccines in the table.

Authors

We thank the reviewer for this comment that allowed us to clarify this point in the manuscript. Monkeypox vaccine recipients were offered other vaccinations for which they were targeted by category (e.g., HAV, HPV) or health condition (if PLHIV), or by age (tetanus). However, the questionnaire did not investigate which type of vaccine it was, so it is not possible to specify the variable further.

DISCUSSION:

Consider adding the following citation and study results into the discussion section of your paper.

Deng L, Lopez LK, Glover C, Cashman P, Reynolds R, Macartney K, Wood N. Short-term Adverse Events Following Immunization With Modified Vaccinia Ankara-Bavarian Nordic (MVA-BN) Vaccine for Mpox. JAMA. 2023 May 5. doi: 10.1001/jama.2023.7683. Epub ahead of print. PMID: 37145654.

Authors

We thank the author for the suggestion of this relevant paper, which was included and commented on in the manuscript discussion.

Reviewer 4 Report

I read the work with interest, I think it is interesting and has a good methodology. I think it deserves publication.

I would like the authors following the results collected to be able to better discuss some points:

- towards which population group is vaccination recommended? given the safety of the administration, could there be further therapeutic indications for other groups of the population?

- there are ethical reasons regarding access to the vaccine perhaps in other more affected countries which should be reported see for example: doi: 10.3390/vaccines9060538.

- the bibliography could be a little expanded

Author Response

Rev4

I read the work with interest, I think it is interesting and has a good methodology. I think it deserves publication.

Authors

We thank the Reviewer for the appreciation of our work and the valuable time spent on their review.

I would like the authors following the results collected to be able to better discuss some points:

- towards which population group is vaccination recommended? given the safety of the administration, could there be further therapeutic indications for other groups of the population?

Authors

We thank the Reviewer for their insight. The categories for which vaccination is offered are, as of now, the same as those for which it was offered during the study period (reported in the Methods section and in the introduction, with the local regulation cited).

Even considering the recent WHO end-of-emergency declaration for monkeypox, we do not think an extension of the vaccination campaign targets is possible at this time.
However, as suggested by the Reviewer, these good tolerability results may help in the future should there be a need to extend vaccination.

- there are ethical reasons regarding access to the vaccine perhaps in other more affected countries which should be reported see for example: doi: 10.3390/vaccines9060538.

Authors

We thank the Reviewer for the suggestion of this interesting paper. Indeed, equitable access to vaccinations is an issue of great importance and certainly deserves to be addressed. In our manuscript, however, we focused only on AEFIs occurred following vaccination; we are currently conducting other studies in which we analyze how the monkeypox vaccine campaign was conducted and whether it was equitable or not in supply. We will consider this important contribution.

- the bibliography could be a little expanded

Authors

We thank the reviewer for this comment. We have added some citations to the bibliography; indeed, the literature currently not rich in original data on this topic. However, we have included a very recent JAMA research letter that analyzes the rates of AEFIs found in a large number of subjects vaccinated with the monkeypox vaccine.

Round 2

Reviewer 2 Report

Thank you for you justification and editions of the paper.

Acceptable

Author Response

The manuscript was revised further following the Editor's advice.  We are then waiting to have the file for proofreading so that we can proceed. 

Reviewer 4 Report

now is publishable

Author Response

(The authors gave the same response as above.)
